# Experimental Study on the Seismic Performance of Recycled Concrete Hollow Block Masonry Walls

**Chao Liu [1,\*], Xiangyun Nong [1], Fengjian Zhang [1], Zonggang Quan [2] and Guoliang Bai [1]**

[1]   College of Science, Xi'an University of Architecture and Technology, Xi'an 710055, China;
    nongxiangyun@xauat.edu.cn (X.N.); zfj9608@163.com (F.Z.); guoliangbai@126.com (G.B.)
[2]   Xi'an Research & Design Institute of Wall and Roof Materials, Xi'an 710055, China; quanzg01@163.com
\*   Correspondence: chaoliu@xauat.edu.cn; Tel.: +86-180-9256-1062

**Abstract:** This paper aims to manufacture recycled concrete hollow block (RCHB) which can be used for the masonry structure with seismic requirements. Five RCHB masonry walls were tested under cyclic loading to evaluate the effect of the axial compression stress, aspect ratio, and the materials of structural columns on the seismic performance. Based on the test results, the failure pattern, hysteresis curves, lateral drift, ductility, stiffness degradation, and the energy dissipation of the specimens were analyzed in detail. The results showed that with the increase of aspect ratios, the ductility of RCHB masonry walls increased, but the horizontal bearing capacity and energy dissipation of RCHB masonry walls decreased. With the increase of compressive stress, the bearing capacity and energy dissipation performance of RCHB masonry walls were improved, and the stiffness degraded slowly. The results also demonstrated that the RCHB masonry walls with structural columns, depending on whether the structural columns were prepared by ordinary concrete or recycled concrete, could increase the bearing capacity, ductility, and energy dissipation of specimens. The research confirmed that RCHB masonry walls could meet the seismic requirements through thoughtful design. Therefore, this study provided a new cleaner production for the utilization of construction waste resources.

**Keywords:** construction and demolition wastes; resource utilization; recycled concrete hollow block; masonry walls; seismic performance

## 1. Introduction

Since the beginning of human civilization, construction and demolition wastes (CDW) had been becoming a global problem that affects the sustainable development of the resources and the environment [1]. To date, more than a hundred billion tons of CDW were generated in the world, while approximately 30% to 50% of them were waste concrete [2]. However, this considerable amount of waste concrete is mainly disposed of in landfills, resulting in a severe environmental problem [3–6]. In order to protect the ecological environment and achieve sustainable development, reducing, reusing, and recycling construction waste is a desirable approach to preserve the ecological environment [7–11]. Therefore, many countries have passed legislation to encourage the recycling of waste concrete for the resource utilization of recycled concrete.

Recycled concrete aggregates (RCA) can be obtained from waste concrete by crushing, screening, cleaning, and separating [12,13]. Previous studies have been conducted to characterize the potential advantages and drawbacks of RCA. Being different from natural aggregates (NA), RCA has a layer of old mortar attached to the surface, which has more loose pores at the interface [14]. The attached old mortar brings with it worse properties: a lower apparent density and higher water absorption [15,16]. Consequently, these properties cause the mechanical properties of RCA to be inferior to that of NA, which limits the application of recycled concrete in civil engineering. Although the mechanical

properties of recycled aggregates are poor, different scholars, institutions, and countries have been applying recycled aggregates into structures through rational design and experiment in recent years, and in some cases, good results have been achieved [17–23].

To further promote the recycling of construction waste, growing studies on the production of recycled concrete blocks prepared by RCA have been conducted in years. Researchers proposed that the mechanical properties of recycled concrete blocks mainly depend on the substitution rate and qualities of RCA. Soutsos et al. [24] pointed out that the increase of recycled fine aggregates content has a more significant impact on the reduction of recycled concrete block strength. Therefore, the maximum replacement rate of recycled fine aggregates is recommended to be 20%. Bai et al. [25] also confirmed the findings through a similar experiment. A study conducted by Poon et al. [26] aimed to investigate the mechanical properties of recycled concrete blocks. They found that the substitution rate of RCA below 50% had little effect on the compressive strength of recycled concrete blocks. However, the compressive strength of recycled concrete hollow block (RCHB) decreases with the increase of the content of low-grade RCA (The content of soil or broken brick in aggregates being >10%) [27]. Guo et al. [28] investigated the influence of different substitution rates of recycled concrete aggregates on the mechanical properties of recycled concrete blocks. The test results illustrated that with the substitution rate up to 75%, the strength of recycled concrete blocks slightly decreases but still complies with the standards. An experiment by Sabai et al. [29] showed that the compressive strength of recycled concrete blocks with 100% RCA can achieve the target of 7 MPa, and even the minimum strength requirement of construction.

Despite the fact that the compressive strength of the concrete block is the critical performance index about whether the block can be used in a masonry structure, the mechanical performance of a masonry prism is more reflective of the actual stress state of the masonry structure. Corinaldesi et al. [30] studied the compressive, shear, and bond strength of recycled mortar prisms, and found that the shear and compressive strength of recycled mortar prisms were close to or even better than that of ordinary mortar prisms. Guo et al. [28] conducted a study of recycled concrete block prisms, and the conclusions are consistent with that of Corinaldesi et al. [30]. In general, the shear strength of the recycled concrete prisms is close to that of the ordinary concrete prisms. However, the seismic performance of the masonry structure, especially the hysteretic characteristics and energy dissipation capacities of the masonry is unclear by the tests of the masonry prisms.

Through the analysis of the above examples, it can be proved that the mechanical properties of recycled blocks can satisfy the requirement of practical application, and the shear properties of masonry assemblages fabricated by recycled concrete blocks are close to those of ordinary masonry assemblages. However, there is still a lack of knowledge about the recycled concrete blocks, and whether they can be used to produce masonry structures with seismic requirements, especially regarding the research about the seismic performance of masonry structures, which is crucial for addressing the utilization problem of RCA. Therefore, if the waste concrete can be recycled to produce RCHB which can be used in the structures with seismic requirements, this type of RCHB will be popularized and further applied on a larger scale.

In view of this, RCA generated from waste concrete was used to produce a new type of RCHB which can be used for masonry structures. In this experiment, three RCHB masonry walls without structural columns constraint, one RCHB wall constrained by the recycled aggregate concrete structural columns, and other walls constrained by ordinary concrete structural columns were manufactured for seismic performance testing. The seismic behavior of the specimens, such as the failure pattern, the hysteresis curves, the skeleton curves, the ductility coefficient, and the energy dissipation of the specimens was analyzed under cyclic loading. The influences of aspect ratio, vertical axial stress, and different materials used for structural columns on the seismic performance of RCHB masonry walls were also studied. Finally, the seismic capacity of RCHB masonry structure under seismic loading was attained.

## 2. Experimental Program

### 2.1. Materials

Ordinary Portland cement with a 28 d strength grade of 42.5 MPa was employed in this experiment. Ordinary mortar with a standard 28 d strength grade of 10 MPa was used as the binder. The fine NA with a grain size of 0–5 mm were river sand. RCA was provided by Shaanxi Jianxin Technology Environmental Protection Co., Ltd. The RCA with a grain size of 5–10 mm (Figure 1) were crushed, cleaned, sieved, and separated from waste concrete. The fine NA and coarse NA were replaced by 0% recycled fine aggregates, and 100% RCA, respectively, and the mixture proportion of recycled aggregate concrete (RAC) is shown in Table 1. The grading of aggregates and properties of aggregates are shown in Figure 2 and Table 2, respectively. Based on the Chinese standard GB/T 41,112,013 [31] and the GB/T 8239-2014 [32], RCHB, prepared by RAC, with a compressive strength of 10 MPa were used for the fabrication of masonry walls, and the dimension of RCHB is 390 mm × 240 mm × 190 mm (Figure 3). A hot-rolled ribbed (HRB335) steel bar with a yielding strength of 335 MPa was used as longitudinal rebars. A hot-rolled plain (HPB235) steel bar with yielding strength of 235 MPa was selected as stirrups.

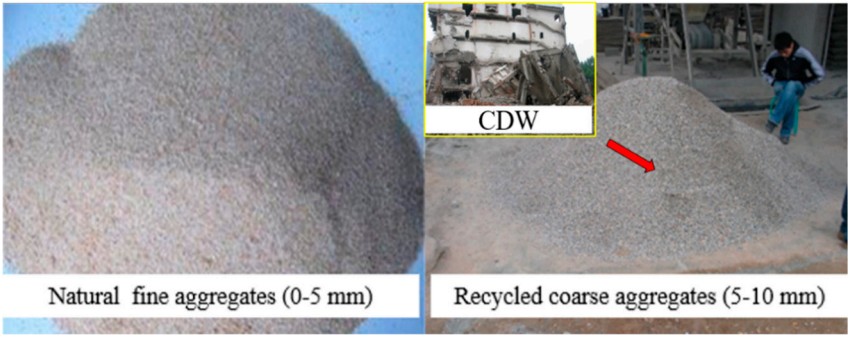

**Figure 1.** Aggregate components.

**Table 1.** Composition of recycled concrete block.

| RCHB Components | Mixture Proportion/kg·m$^{-3}$ |
|---|---|
| Water | 150 |
| Cement | 375 |
| Recycled coarse aggregates | 945 |
| Recycled fine aggregates | — |
| Natural coarse aggregates | — |
| Natural fine aggregates | 630 |

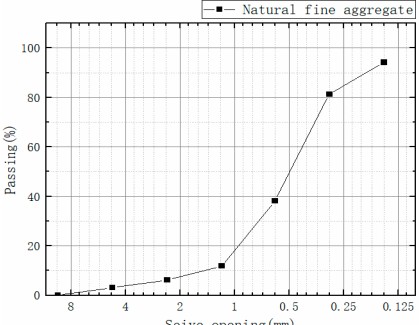
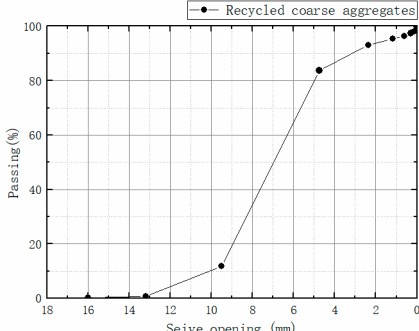

**Figure 2.** Fuller grading curves.

**Table 2.** Properties of aggregates.

| Aggregate | Grading /mm | Apparent Density /kg·m$^{-3}$ | Bulk Density /kg·m$^{-3}$ | Water Absorption/% | Crushing Index | Fineness Modulus |
| --- | --- | --- | --- | --- | --- | --- |
| NA | <5 | 2724.8 | 1448.4 | 1.02 | — | 2.18 |
| RCA | 5–10 | 2521.7 | 1096.1 | 3.50 | 24.74 | — |

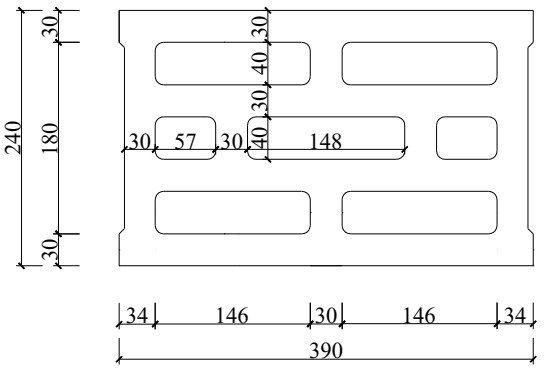 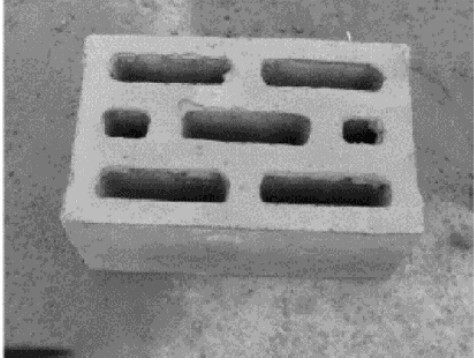

**Figure 3.** The dimension of recycled concrete hollow block.

## 2.2. Design of Specimens

Figure 4 shows the masonry procedure of all the specimens, including masonry the bottom beams, infill walls, structural columns, and ring beams. It is important to mention that ordinary mortar with a strength of 10 MPa was selected to masonry the mortar joints of the infill walls, and the fullness degree of mortar joints should be up to 80% to ensure the propagation of shear forces according to the seismic design requirements. In addition, the horizontal joint width was required to range from 8 to 12 mm.

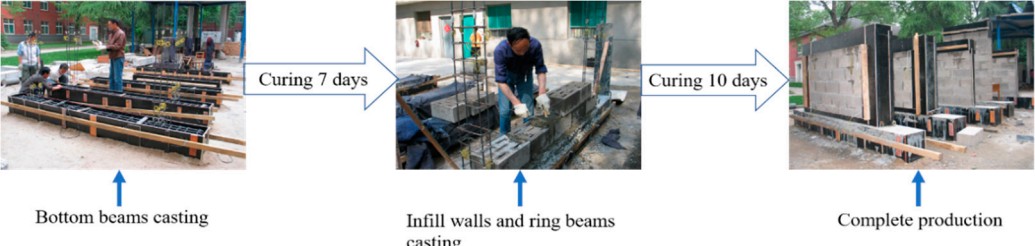

**Figure 4.** Process of the construction of masonry wall specimens.

The experimental parameters used to determine the seismic performance of specimens were the compressive stress, aspect ratios, and the materials of the structural columns. The test specimens were divided into two groups. The first group consisted of three walls W1, W2, and W3, and the cyclic loading tests of them were conducted to analyze the influence of different compressive stress and aspect ratio on the seismic capacity. The second group contained two specimens, W4 and W5, which were constrained by structural columns, and the tests of them were conducted to analyze the influence of structural columns of different materials on the seismic capacity. The other components of all specimens, such as bottom beams and ring beams, were consistent in the test. Additionally, the value of the compressive stress was 0.6 MPa and 0.9 MPa, respectively, corresponding to the compressive stress of the upper weight on the middle floor and the bottom floor of an ordinary 9-storey residential building, respectively. Properties and dimensions of the five masonry walls are shown in Table 3 and Figure 5.

**Table 3.** Design of recycled concrete hollow block (RCHB) masonry walls.

| No. | W1 | W2 | W3 | W4 | W5 | Unit |
|---|---|---|---|---|---|---|
| Width | 2000 | 2000 | 1200 | 2240 | 2240 | mm |
| Height | 1400 | 1400 | 2000 | 1350 | 1350 | mm |
| Aspect ratio | 0.700 | 0.700 | 1.667 | 0.603 | 0.603 | — |
| Vertical load | 288 | 432 | 259 | 323 | 323 | kN |
| Vertical Axial stress | 0.6 | 0.9 | 0.9 | 0.6 | 0.6 | MPa |
| Materials of Structural columns | — | — | — | NA | RCA | — |

Key: 2Φ10=2 bars of 335MPa

Φ6@200= 6mm stirrup bars of 300MPa with 200mm spacing

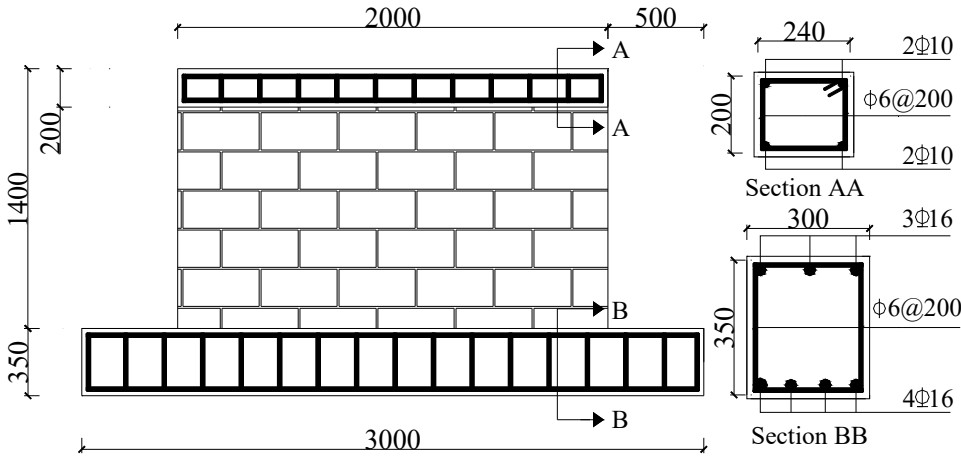

(**a**) W1 and W2.

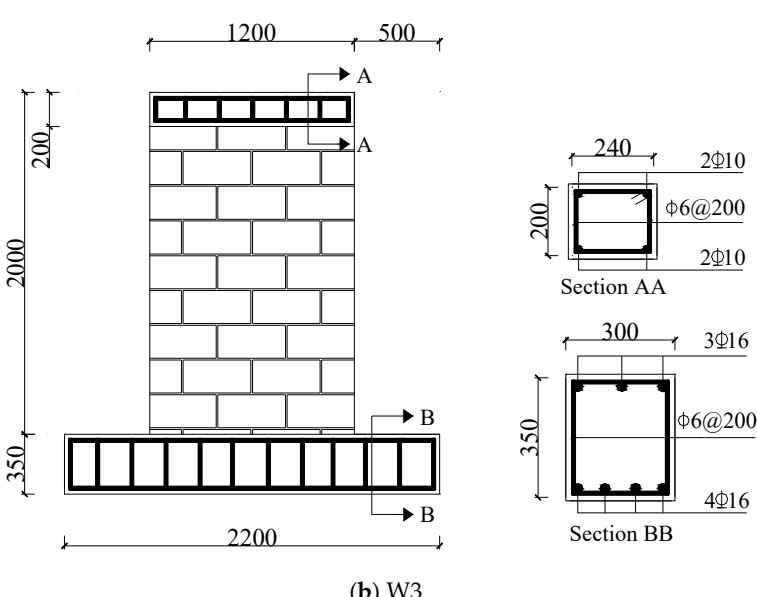

(**b**) W3

**Figure 5.** *Cont.*

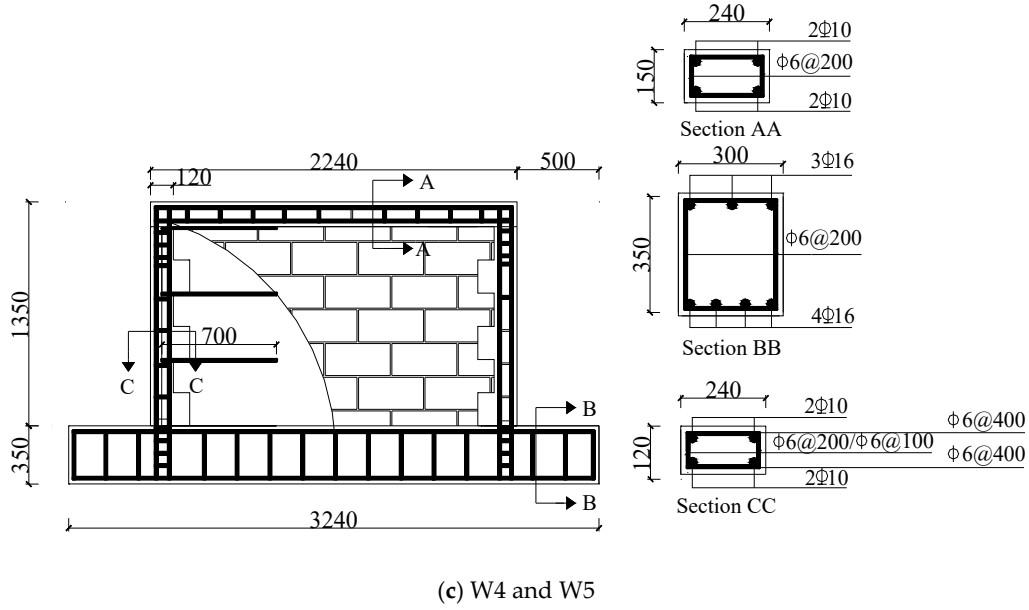

(**c**) W4 and W5

**Figure 5.** Size of the specimens: (**a**) W1 and W2; (**b**) W3; (**c**) W4 and W5.

The dimensions and the steel arrangement of the specimens should be meeting the specifications as outlined in GB/T 41112013. The section size of the structural column, ring beam, and bottom beam were 240 mm × 120 mm, 240 mm × 200 mm (or 240 mm × 150 mm) and 300 mm × 350 mm, respectively. Ordinary Portland concrete with 28 d axial strength of 42.5 MPa was used in the bottom beams during the test. The reinforcement arrangement is shown in Figure 5.

### 2.3. Test Procedure

The loading procedure included two steps: a load-controlled step and a drift-controlled step, which can be observed in Figure 6. First, the specimens were subjected to a vertical pre-compression load, which was kept constant during each test (Figure 2). After 20 min, 20 kN lateral loads were applied to the specimens in advance and repeated three times. Before formal loading, the experiment instruments were checked to see if they work properly, and then the experiment was conducted. During the load-controlled stage, the force control loading was adopted in the experiment. During this time, the horizontal load was applied cyclically with an increment of 40 kN until the walls cracked. After the walls initially cracked, the application method was changed to the drift control step. Then, the displacement levels of the cracked point were regarded as the first displacement step and set as the increment for the subsequent displacement cycles. Cyclic loading was applied twice at each displacement magnitude until wall failure. Subsequently, the tests were terminated when the lateral load of specimens decreased to about 85% of its peak value, or the number of specimen cracks reached about 50% of the total number of masonry mortar joints.

### 2.4. Test Device and Measuring Arrangement

The test device and the measuring instrument of deformation are presented in Figure 7. The layout of the measuring instruments and data collections were introduced as follows: (1) A horizontal linear variable differential transformer (LVDT) was placed in the middle of the ring beam to measure the horizontal displacement of the wall; (2) force sensors continuously recorded the values of vertical load and horizontal load applied by the vertical jack and the horizontal actuator; (3) the status of cracks was observed by the naked eye during experiments during the test. The occurrence, development, width of cracks, and maximum crack width were continually recorded at every loading process, and a marking pen described the shape of the cracks. The number and the values of cycles were also recorded; (4) two vertical LVDTs were placed on both sides of the bottom beam respectively to measure

vertical displacement. A horizontal LVDT was arranged at one side of the bottom beams to measure horizontal displacement.

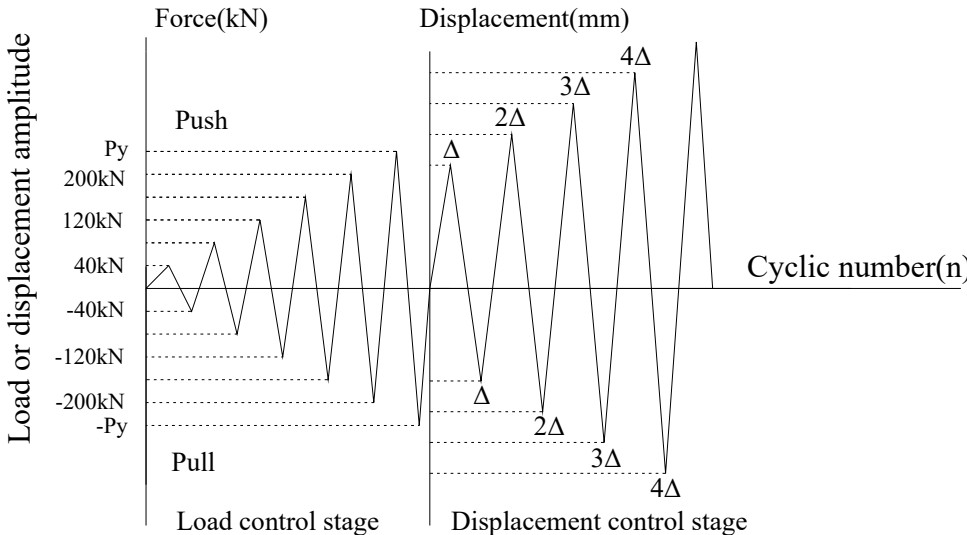

**Figure 6.** Lateral loading process.

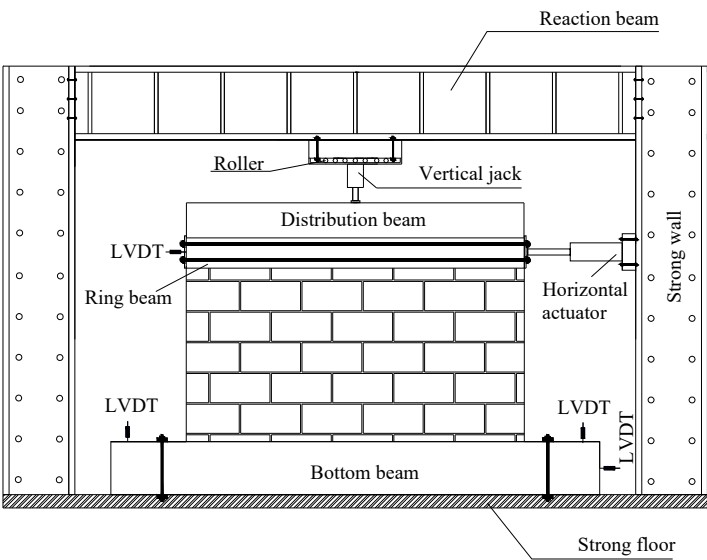

**Figure 7.** Schematic representation of the test setup.

## 3. Test Results

### 3.1. Failure Phenomenon and Pattern

The failure patterns of the five specimens are presented in Figure 8. All the specimens experienced three phases, such as cracking, peak, and failure in the experiment, but the failure process of the specimens W1, W2, and W3 were different from the specimens W4 and W5. Without structural columns, the specimens W1, W2, and W3 (bare walls) reached the ultimate strength and were destroyed immediately after the walls cracked, indicating the specimens had poor ductility and bearing capacity. On the contrary, the specimens W4 and W5 (reinforced walls) reflected better ductility and bearing capacity due to the constraints of the structural columns and ring beams. The analysis of the failure patterns of all the specimens is as follows:

1.  In the initial stage of loading, horizontal cracks occurred on both sides of the wall root of the specimen W1 due to the flexural effect. By this time, no obvious cracks occurred in the diagonal

direction because the shear strength of the diagonal direction did not reach the cracked load. As the horizontal load increased, the horizontal cracks propagated rapidly and linked together gradually at the wall root, and the visible flexural effect was more apparent under horizontal loading. When the horizontal load reached the ultimate load, the mortar joint between the infill wall and the bottom beam was destroyed completely. Meanwhile, the specimen hardly bore the lateral load, indicating that specimen W1 failed. Therefore, the failure process of W1 was predominantly affected by the flexural effect;

2.  During the initial loading stage, small horizontal cracks gradually appeared between the horizontal mortar cushion and the bottom beam of the specimen W2. As the horizontal load increased, the horizontal cracks of the wall root appeared along the ladder joints from the lower to the upper. When the lateral loads reached 80% of its ultimate loads, the cracks of the ladder joints continually widened and propagated simultaneously towards the diagonal direction. At this moment, numerous cracked blocks were spalled from the diagonal of the wall. When the lateral load increased to the ultimate load, the previously formed cracks at the diagonal direction linked together, and the bearing capacity of the specimen W2 declined rapidly, indicating that the specimen was destroyed;

3.  Due to the large aspect ratio, the flexural influence on W3 was evident under the horizontal load, and the failure pattern of W3 was similar to W1;

4.  In the initial loading stage, small stepped cracks appeared along the wall root and the vertical joints of the infill walls of the specimens W4 and W5. As the horizontal load increased, tiny horizontal cracks gradually appeared on both sides of the bottom of the structural columns, and other small cracks appeared at the upper oblique of the structural columns. As the horizontal load further increased, these cracks gradually widened and nearly connected, tearing apart the RCHB of the diagonal direction. When the lateral loads neared the ultimate loads, the diagonal cracks and horizontal cracks widened rapidly and linked together to form a large "X" shape crack under cyclic loading. At the same time, the spalling of the cracked blocks was observed in the diagonal direction of the specimens. Subsequently, the bearing capacity of the specimens reduced rapidly to 15% of its maximum values, indicating that the specimens W4 and W5 were destroyed.

### 3.2. Characteristic Load and Displacement

As can be noticed in Table 4, the ultimate load and the cracking load of the specimens W1 was 98% and 185% lower than that of W2, respectively, which showed that increasing the vertical load could increase the bearing capacity of specimens. The cracking load and the ultimate load of the specimen W2 was higher compared to W3, which indicated that the bearing capacity of specimens was decreased with the increasing aspect ratio.

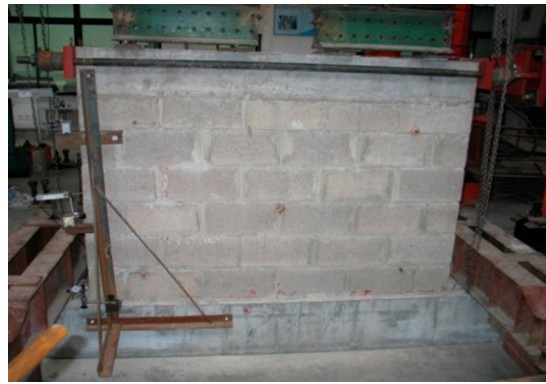
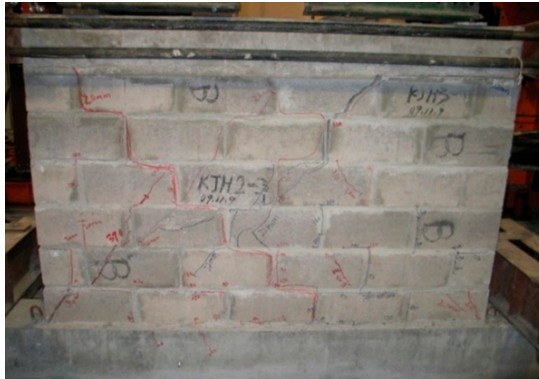

(**a**) W1                                          (**b**) W2

**Figure 8.** *Cont.*

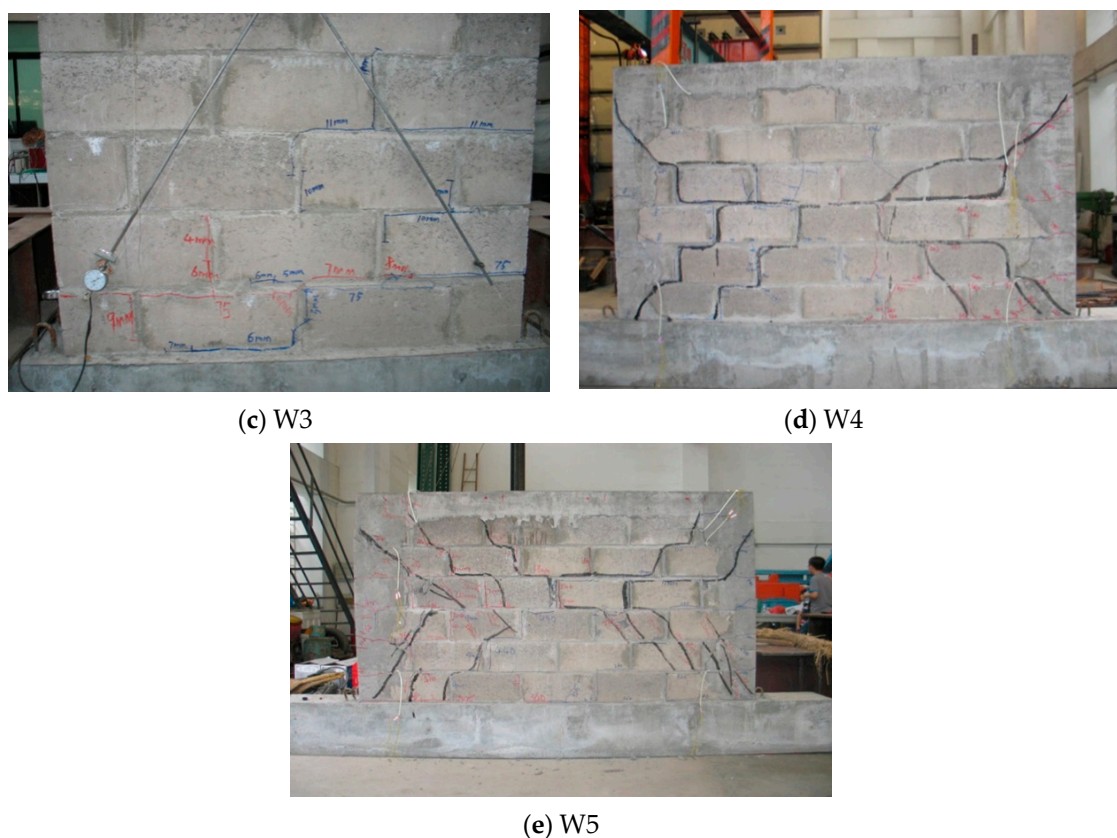

(**c**) W3            (**d**) W4

(**e**) W5

**Figure 8.** Failure mode of specimens: (**a**) W1; (**b**) W2; (**c**) W3; (**d**) W4; (**e**) W5.

**Table 4.** Characteristic load and displacement.

| No. | W1 | W2 | W3 | W4 | W5 | Unit |
|-----|------|------|------|------|------|------|
| $P_{cr}$ | 136.5 | 389.7 | 76.7 | 309.9 | 234.8 | kN |
| $\Delta_{cr}$ | 2.49 | 13.60 | 4.19 | 0.80 | 0.83 | mm |
| $P_u$ | 201.58 | 399.30 | 108.10 | 457.99 | 431.80 | kN |
| $\Delta_u$ | 8.995 | 14.905 | 20.990 | 6.960 | 2.970 | mm |
| $P_{cr}/P_u$ | 0.677 | 0.975 | 0.709 | 0.676 | 0.543 | — |

Key: $P_{cr}$ is the cracking point; $\Delta_{cr}$ is the cracking displacement; $P_u$ is the peak load; $\Delta_u$ is the peak displacement.

The ultimate load and the cracking load of W4 and W5 exhibited about a 127%, 127%, 72%, and 114% increase compared with W1, respectively, indicating that the constraint imposed by the structural columns could significantly improve the bearing capacity of the walls. Besides that, the cracking load and ultimate load of the W5 were slightly lower than that of the W4, reflecting the bearing capacity of the ordinary concrete structural column was higher than that of the RAC structural column.

## 4. Test Analysis

### 4.1. Hysteresis Behaviour

The recorded hysteresis curves of all the specimens under cyclic loading are presented in Figure 9. Based on Figure 9a–c, the hysteresis curves of the specimens W1, W2, and W3 are considerably linear in the initial elastic stage. After the walls cracked, the area of the three hysteresis curves slightly increased, and the shapes of the hysteresis curves transformed from the original shuttle to the reversed "S" shape, showing a partial pinching effect. The area of the hysteretic curves of W1 and W3 nearly stayed the same due to the flexural effect, and the profile of the two hysteresis curves remained stable until the specimens failed. In addition, the area of the three hysteresis curves showed a marked difference.

The area of the hysteresis loop of W2 was the best, followed by that of W1, while W3 was the worst. For W1 and W2, the specimen W2 with higher compression stress had a better energy dissipation compared to W1, indicating that the increase in compressive stress can improve the energy dissipation capacity of specimens. However, for W2 and W3, specimen W3, with a higher aspect ratio, had a worse energy dissipation compared with W2, indicating that the increase in aspect ratio can reduce the energy dissipation capacity of specimens.

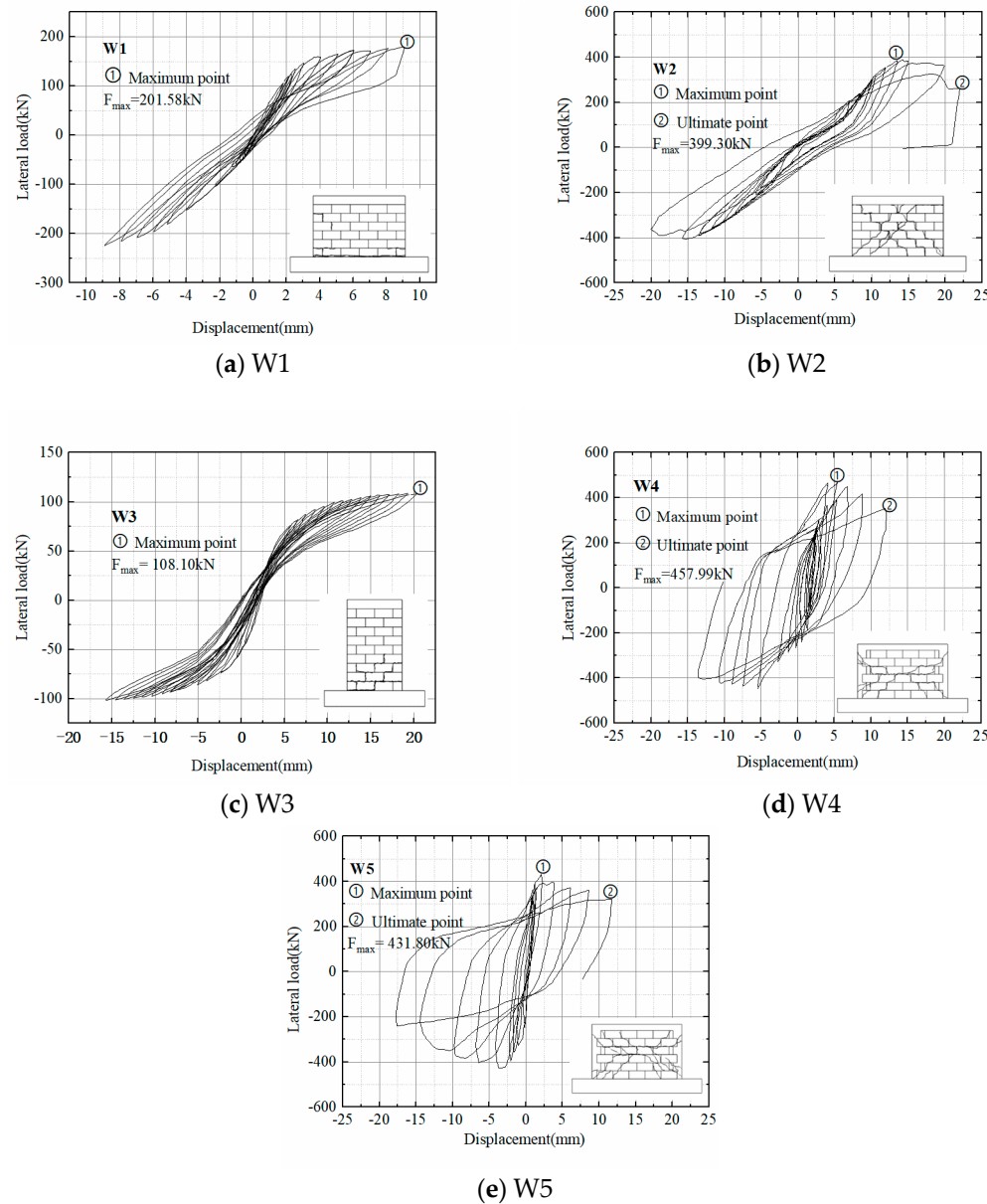

**Figure 9.** Hysteretic curves of five specimens: (**a**) W1; (**b**) W2; (**c**) W3; (**d**) W4; (**e**) W5.

As shown in Figure 9d,e, the hysteresis loops of W4 and W5 have a similar linear relationship in the initial elastic stage. During this time, the horizontal drift of the top walls and residual deformation were small after unloading. The area of the hysteresis loops of the specimens was small and nearly overlapped. After the walls cracked and entered into a plastic stage, especially after the maximum load, the hysteresis loops showed partial pinch effect, and the area of hysteresis loops was significantly increased, which indicated that the structural columns had an apparent energy dissipation capacity. Compared with W4, the maximum bearing capacity of W5 was slightly lower than that of W4, which

indicated that the increasing bearing capacity of RAC structural columns was lower than that of ordinary concrete structural columns.

## 4.2. Skeleton Curve

The skeleton curve is the envelope curve obtained by connecting the peak points of the P–D hysteretic loop of the first cycle in each loading stage, mainly reflecting the cracking load and ductility of the wall.

As can be noticed in Figure 10a, before the specimen crack, the three skeleton curves of the specimens are a straight line in the elastic stage. After the specimens cracked, the three skeleton curves started becoming nonlinear in the plastic stage. When the lateral load closed to the maximum load, the skeleton curves gradually tilted toward the axis with increasing lateral drift. Besides that, the three skeleton curves show an apparent difference. Comparing the specimens W1 and W2, the maximum load of W2 was higher than that of W1, and the specimen W2 had a steeper degraded section after cracking, indicating that increasing the compression stress can increase the bearing capacity and decrease the ductility of specimens. The cracked load and maximum load of W3 were lower compared to W2, and the horizontal section of W3 was longer after cracking, which indicated that the increasing aspect ratio could reduce the bearing capacity and increase the ductility.

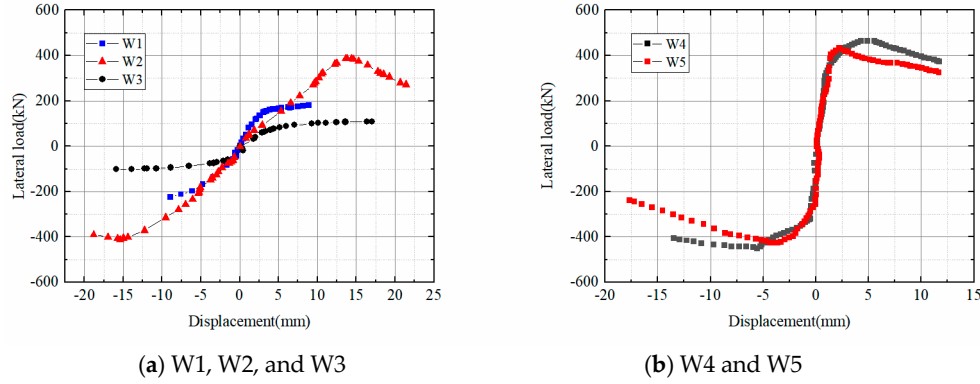

(**a**) W1, W2, and W3          (**b**) W4 and W5

**Figure 10.** Skeleton curves of specimens: (**a**) W1, W2, and W3; (**b**) W4 and W5.

It can be seen from Figure 10b, the bearing capacity and the deformation property of the specimens W4 and W5 are obviously higher than that of W1, W2, and W3. The stiffness degrades of the two specimens W4 and W5 was similar before cracking. After cracking, the two skeleton curves showed obvious differences in the elastic-plastic and failure phases. The maximum load of the specimen W4 was higher in comparison to the specimen W5, and the specimen W4 had a relatively smooth descending section, which reflected that the specimen W4 with ordinary concrete structural columns had the better bearing capacity and ductility than that of the specimen W5 with RAC structural columns.

## 4.3. Ductility

The ductility coefficient and shift angle are important parameters to assess the seismic performance of the structure, and they also are the essential characteristics to evaluate the deformation ability of the specimen.

Nowadays, the calculation methods for ductility are different, and each method has its characteristics. In this paper, the displacement ductility coefficient ($\mu = \frac{\Delta_u}{\Delta_y}$) is used for representing the ductility, where $\Delta_y$ is the yield displacement, which is obtained by using the equivalent energy method [33], $\Delta_u$ is the displacement corresponding to the lateral load of 85% of the ultimate load. The Shift angle is calculated as $\theta = \frac{\Delta_u}{H}$, where H is the distance from the top surface of the bottom beam to the lateral loading point. The ductility coefficient and shift angle of each wall are shown in Table 5.

**Table 5.** Ductility coefficients and shifts angle of specimens.

| No. | W1 | W2 | W3 | W4 | W5 | Unit |
|-----|------|--------|--------|-------|-------|------|
| $\Delta_y$ | 4.333 | 12.510 | 6.721 | 1.416 | 0.926 | mm |
| $\Delta_u$ | 8.995 | 14.905 | 20.990 | 6.960 | 2.970 | mm |
| H | 1200 | 1200 | 1800 | 1200 | 1200 | mm |
| $\mu$ | 2.075 | 1.191 | 3.123 | 4.915 | 3.207 | — |
| $\theta$ | 0.007 | 0.012 | 0.011 | 0.006 | 0.002 | — |

As can be seen from Table 5, the mean values of ductility coefficient $\mu$ of bare walls are between 1.191 and 3.123, and the mean values of ductility coefficient of W2 are smallest. Compared to W2, the increase in ductility was 42.6% and 162.2% for the specimens W1 and W3, indicating that increasing the compression stress could decrease the ductility of specimens, but increasing the aspect ratio could increase the ductility of specimens.

The ductility coefficients of reinforced walls were higher than 3, meeting the specification requirements of GB 50003-2011 [34] ($\mu \geq 3$), which indicated that the structural columns enhanced the deformation ability of both walls, and decreased the brittleness of specimens. According to the results of Table 5, the ductility coefficient and the lateral shift angle of W4 were higher than that of W5, which reflected that the ductility of the ordinary concrete structural columns was better than that of the RAC structural columns.

*4.4. Stiffness Degradation*

The stiffness degradation factor ($K_i$) is expressed as follows:

$$K_i = \frac{|P_i| + |-P_i|}{|\Delta_i| + |-\Delta_i|},$$

In which $P_i$ is the maximum load and $\Delta_i$ is the corresponding displacement. The stiffness degradation of all specimens is shown in Figure 11.

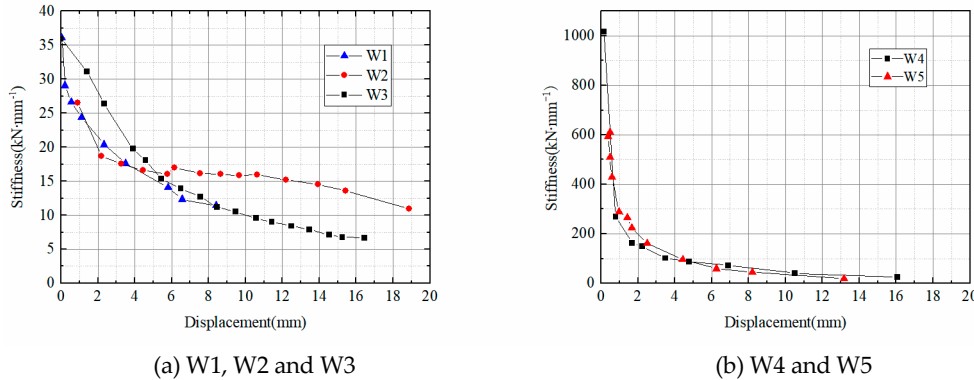

(a) W1, W2 and W3        (b) W4 and W5

**Figure 11.** Stiffness deterioration curves of specimens: (**a**) W1, W2, and W3; (**b**) W4 and W5.

The stiffness degradation tendency of all the specimens is plotted in Figure 11. This figure shows that the stiffness of bare walls gradually decreases with the increase of the displacement. After the displacement reached the ultimate displacement, the stiffness of bare walls degraded quickly due to the accumulation of cracks. It can be easily observed that the stiffness degradation rate of W2 was lower than that of W1 and W3 after cracking. In comparison with W1, the stiffness of W2 generally degraded, which indicated that the increase of vertical compression could reduce the degradation rate of the stiffness of specimens. In addition, when other factors were equal, a comparison of the stiffness degradation rate of W2 and W3 showed that the aspect ratio had a negative influence on the degradation rate of the stiffness of specimens.

It can be seen from Figure 11b that the stiffness degradation tendency of W4 nearly coincides with that of W5. In the initial stage, the stiffness of the specimens decreases quickly with the increase of the displacement. After the displacement reached the ultimate displacement, the stiffness degradation of reinforced specimens tended to be more smooth. The analysis stated that the effective constraint applied by structural columns could significantly alleviate stiffness degradation. In comparison with W5, the stiffness of W4 deteriorated rapidly at an early stage and deteriorated slowly at a late stage, but in general, the stiffness curves of W4 and W5 were nearly overlapping, indicating that the stiffness degradation of the specimen with RAC structural columns was close to that of ordinary concrete structural columns.

Table 6 lists the secant stiffness of characteristic points. It can be found that the stiffness degradation rate of W4 was higher than that of W5 before cracking, and after cracking, the stiffness degradation rate of W4 was slightly lower than that of W5.

**Table 6.** Secant stiffness of all the specimens at characteristic points.

| Specimen | Secant Stiffness (kN·mm$^{-1}$) | | |
| --- | --- | --- | --- |
| | Initial Point | Cracking Point | Ultimate Point |
| W1 | 79.38 | 52.53 | 21.95 |
| W2 | 60.67 | 44.51 | 14.12 |
| W3 | 35.95 | 24.76 | 6.04 |
| W4 | 1017.44 | 234.25 | 25.26 |
| W5 | 611.59 | 251.24 | 22.53 |

### 4.5. Energy Dissipation Capacity

Energy dissipation performance refers to the energy dissipation capacity of the wall under the action of the earthquake load, which is obtained by calculating the area enclosed by the overall hysteresis loop of the first loading cycle. It is an important indicator to measure the seismic performance of the structure. The curves of energy dissipation per cycle are presented in Figure 12.

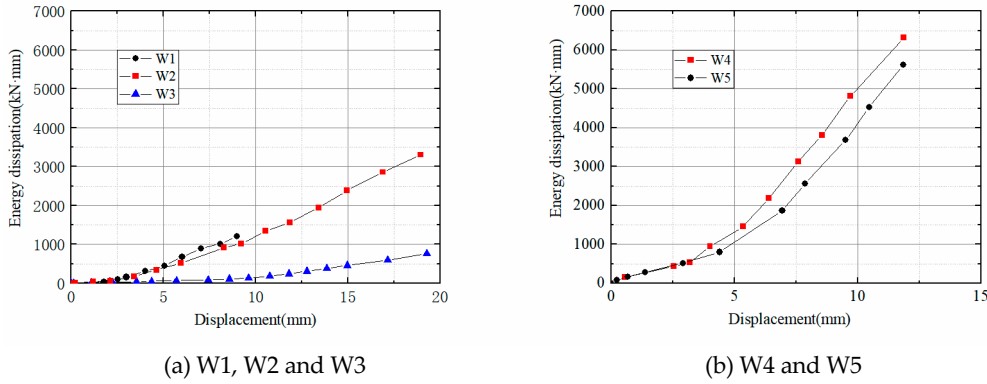

(a) W1, W2 and W3 (b) W4 and W5

**Figure 12.** Energy dissipation per cycle for all test specimens: (**a**) W1, W2, and W3; (**b**) W4 and W5.

As shown in Figure 12a, the energy dissipation increases with the increasing lateral displacement, and the energy dissipation of W2 was the best, then the next was W1, and the worst was W3. Comparing of W1 and W2, the energy dissipation capacity of W2 was higher, indicating that, within a certain range, the vertical compression had a positive influence on the energy dissipation capacity of the specimens. The energy dissipation of W3 was lower compared with W2, indicating that increasing the aspect ratio could decrease the energy dissipation capacity of specimens.

The energy dissipation curves of the specimens W4 and W5 are shown in Figure 12b. The specimens W4 and W5 were found to have a higher energy dissipation in comparison with the bare walls, indicating that the structural columns could significantly increase the energy dissipation capacity of walls.

To further quantify the hysteretic energy dissipation performance of the specimens, the equivalent viscous damping coefficient is used to measure the energy dissipation capacity of the structure under earthquake resistance. $h_{e,u}$ and $h_{e,f}$ are the equivalent viscous damping coefficients corresponding to the maximum load and the ultimate load, respectively. The equivalent viscous damping can be calculated as follows:

$$h_e = \frac{1}{2\pi} \cdot \frac{S_{ABC}}{S_{\triangle OBD}}$$

where $S_{\triangle BOD}$ and $S_{ABC}$ are the areas enclosed by the shaded hysteresis loop in Figure 13, respectively.

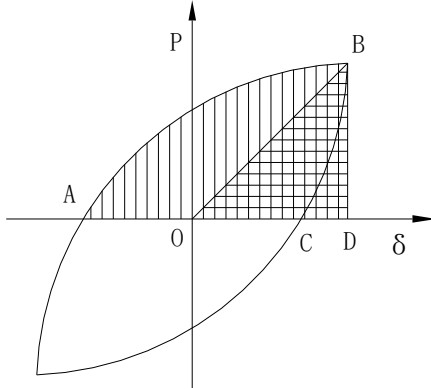

**Figure 13.** Calculated graphics of the equivalent viscous damping coefficient.

The value of the equivalent viscous damping coefficient and energy dissipation per cycle can be observed from Table 7 and Figure 13, respectively. In comparison with W1, the value of $h_{e,u}$ of W2 was 18.6% higher than that of W1, which indicated that the vertical compression could increase the energy dissipation performance of specimens. In terms of W2 and W3, the value of $h_{e,u}$ of W3 was 54.8% lower than that of W2, which illustrated that increasing the aspect ratio had a negative influence on the energy dissipation performance of specimens. It should be noted that W1 and W3 were almost destroyed after reaching the ultimate load, and the hysteresis curve of W2 was not closed when the specimens were destroyed. Therefore, the value of $h_{e,f}$ of W1, W2, and W3 was ignored.

**Table 7.** Equivalent viscous damping coefficients of all specimens.

| No. | W1 | W2 | W3 | W4 | W5 |
|---|---|---|---|---|---|
| $h_{e,\mathrm{u}}$ | 0.087895 | 0.104263 | 0.047126 | 0.240740 | 0.210634 |
| $h_{e,\mathrm{f}}$ | — | — | — | 0.312500 | 0.285784 |

It can be easily found that the value of equivalent viscous damping ratio of W4 and W5 were more significant than that of bare walls, which illustrated that, whether each of the specimens were constrained by ordinary concrete structural columns or RAC structural columns, the equivalent viscous damping ratio and the energy dissipation of the walls were improved. Compared to W4, the energy dissipation viscous damping ratio of W5 exhibited a 12.5% and 8.9% increase at maximum load and ultimate load, which illustrated that the energy dissipation capacity of ordinary concrete structural columns was close to that of RAC structural columns.

## 5. Conclusions

This paper aims to develop a new concrete hollow block with RCA for the structural element. A laboratory test was carried out to investigate the seismic performance of RCHB masonry walls. Based on the experiment data and discussion, the following conclusions can be drawn:

1.  In this paper, waste concrete of construction waste is re-utilized as the resource in manufacturing RCHB. This block can be used in masonry structures with seismic requirements according to the

Chinese standard. The test results indicate that this RCHB masonry structure complies with the seismic requirement through rational design. The above work provides a new solution for the recycling of construction waste;

2. RCHB masonry walls with structural columns, whether the structural columns are ordinary concrete or recycled concrete, can effectively form a constraint on the wall. The resulting constraints improve the bearing capacity and energy dissipation properties of RCHB masonry walls by more than 50%;

3. The ductility of RCHB masonry bare walls is lower than that of the RCHB masonry reinforced wall, indicating that the structural columns can significantly increase the ductility of specimens. The degradation trends of the stiffness of all bare walls are similar. Higher compressive stress can slightly increase the initial stiffness of RCHB masonry walls but accelerates the stiffness degradation after the specimens yielded. The stiffness degradation of all reinforced walls is similar;

4. With the increase of the vertical compressive stress, the bearing capacity and energy dissipation properties of the walls are improved, but the ductility of the recycled concrete block wall is decreased. With the increase of the aspect ratio, the ductility of the wall decreased, but the bearing capacity and the energy dissipation performance of specimens decreased;

5. Although the seismic performance of the RAC structural column is slightly inferior to that of the ordinary concrete structural column, the application of them on the masonry walls can significantly improve the strength and energy dissipation capacity of the RCHB masonry structure. Moreover, compared with the ordinary concrete structural column, the RAC structural column is more economical and environmentally friendly.

**Author Contributions:** Writing—review and editing, C.L. and X.N.; Resources, F.Z., Z.Q. and G.B.

**Funding:** This research was funded by [the Natural Science Foundation of China] grant number [51878546], [the Shaanxi Province Innovative Talent Promotion Plan Project] grant number [2018KJXX-056], [the Shaanxi Provincial Science and Technology Innovation Planning Project] grant number [2016KTZDSF04-02-01], the Shaanxi Provincial Key Research and Development Program] grant number [2018ZDCXL-SF-03-03-02], [the Shaanxi Provincial Key Technology and Science Innovation Team Program] grant number [2016-KCT31], [the Shaanxi Science and Technology Innovation Base] grant number [2017KTPT-19].

**Acknowledgments:** The support from the following fund projects is highly acknowledged: (1) the National Natural Science Foundation of China (51878546); (2) the Shaanxi Provincial Innovative Talents Promotion Program (2018KJXX-056); (3) the Shaanxi Provincial Science and Technology Innovation Planning Project (2016KTZDSF04-02-01); (4) the Shaanxi Provincial Key Research and Development Project (2018ZDCXL-SF-03-03-02); (5) the Shaanxi Provincial Key Technology and Science Innovation Team Program (2016-KCT31); and (6) the Shaanxi Science and technology innovation base "technology innovation Service Platform of solid waste resources energy-saving wall materials science" (2017KTPT-19).

**Conflicts of Interest:** The authors declare no conflict of interest.

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
