# Peer review of "Experimental Study on the Seismic Performance of Recycled Concrete Hollow Block Masonry Walls"

_applsci, doi:10.3390/app9204336_

Round 1

Reviewer 1 Report

This paper addressed the seismic performance of recycled concrete hollow block masonry walls. Applying cyclic loading to evaluate the effect of the axial compression stress, aspect ratio, and the materials of structural columns on the seismic performance is considered the major contribution of this paper. The description of the test is clear for understanding and literature review is good enough. However, this paper in its current state has some question should be addressed:

1. The paper did not evaluate and discuss the effect of the structural columns which use different materials, NA and RCA.

2. The source of RCA and its composition should be stated.

Author Response

Response to Reviewer 1 Comments

Thanks to the reviewers and editors for reviewing the manuscript during the busy schedule and providing valuable comments.

All the changes in the article are marked with bold and underline (E.g. changes).

Point 1: The paper did not evaluate and discuss the effect of the structural columns which use different materials, NA and RCA. 

Response 1: Based on the experimental results, the comparison of the seismic performance of specimen with ordinary concrete structural columns and with recycled aggregate concrete (RAC) structural columns, which used NA and RCA respectively, are added to the article. The changes are as follows, and the rest of the changes (identified as bold and underlined) can be found in the section 4.1, section 4.2, section 4.3, section 4.4, and section 4.5 of manuscript, respectively.

Compared with W4, the maximum bearing capacity of W5 was slightly lower than that of W4, which indicated that the increasing bearing capacity of RAC structural columns was lower than that of ordinary concrete structural columns.

“The maximum load of the specimen W4 was higher in comparison to the specimen W5, and the specimen W4 had a relatively smooth descending section, which reflected that the specimen W4 with ordinary concrete structural columns had the better bearing capacity and ductility than that of specimen W5 with RAC structural columns.”

“According to the results of Table 5, the ductility coefficient and the lateral shift angle of W4 were higher than that of W5, which reflected that the ductility of the ordinary concrete structural columns was better than that of the RAC structural columns.”

“In comparison with W5, the stiffness of W4 deteriorated rapidly at an early stage and deteriorated slowly at a late stage, but in general, the stiffness curves of W4 and W5 were nearly overlap, indicating that the stiffness degradation of the specimen with RAC structural columns was close to that with ordinary concrete structural columns”

“Compared with W4, the energy dissipation viscous damping ratio of W5 exhibited a 12.5% and 8.9% increase at maximum load and ultimate load, which illustrated that the energy dissipation capacity of RAC structural columns was close to that of ordinary concrete structural columns.”

Point 2: The source of RCA and its composition should be stated.

Response 2: Recycled concrete aggregate (RCA) is provided by Shaanxi Jianxin Technology Environmental Protection Co., Ltd. RCA can be obtained from waste concrete by crushing, screening, cleaning, and separating (Xiao, J.Z. Recycled concrete. Beijing: China Architecture &Building Press. Beijing. China.). Compared with natural aggregates (NA), RCA has a layer of old mortar attached to the old aggregate surface (Etxeberria, M.; Vázquez, E.; Marí, A.R. Microstructure analysis of hardened recycled aggregate concrete. Mag. Concr. Res. 2006, 586, 83–90.). This layer of old mortar is hard to peel off, and the content of old mortar in different RCA varies greatly.

Reviewer 2 Report

Dear Authors,

The paper Experimental study on the seismic performance of recycled concrete hollow block masonry walls by Chao Liu, Xiangyun Nong, Fengjian Zhang, Zonggang Quan, Guoliang Bai  is well suited for journal Applied Sciences. The authors tests possibility of application of recycled concrete hollow block for the masonry structure with seismic requirements. The possibility of practical application of the tested blocks is very important, especially in the context of its lower environmental footprint.

The paper is interesting and scientifically valuable. The paper contains parts in good order: introduction, experimental program, results, experiment analysis, conclusions. Unfortunately, the article does not contain a typical chapter “discussion” but it can be considered that this function is fulfilled by chapter “Experiment analysis”. The authors cited corrected references, however there is not a single citation of the journal Applied Sciences.

The article was written enough well in English, is understandable for a reviewer, a person who does not speak English as a mother tongue.

For the reasons stated, I support publication of the paper in the journal after revision (needs minor modification before considering for publication).

Still, the following changes are suggested to improve the quality of the paper:

Major shortcomings:

- line 13 - in the reviewer’s opinion instead off “recycled concrete block hollow” should be “recycled concrete hollow block”. This is stated in the article title and keywords, the abbreviation RCHB corresponds to that name, it is linguistically more correct.

Minor shortcomings:

- line 95 - in the reviewer’s opinion instead off “Experimental programs” will be better “Experimental program”.

- line 113 - in the reviewer’s opinion instead off “The dimension of recycled concrete block” will be better “The dimension of recycled concrete hollow block”.

- line 126 - in the reviewer’s opinion instead off “Process of masonry walls” will be better “Process of construction of masonry wall specimens”.

- line 152 - in the reviewer’s opinion instead off “Test program” will be better “Test procedure”.

- line 240 - in the reviewer’s opinion instead off “Experiment analysis” will be better “Test analysis”.

- line 323, fig. 10b - why the range for horizontal axis ends with 25? Data is only on a part of the chart, the scale of the axis should be adjust (-20,15).

- line 383, fig. 12a - why the range for horizontal axis ends with 25? Data is only on a part of the chart, the scale of the axis should be adjust (-0,20).

- line 383, fig. 12b - why the range for horizontal axis ends with 25? Data is only on a part of the chart, the scale of the axis should be adjust (-0,15).

Author Response

Response to Reviewer 2 Comments

Thanks to the reviewers and editors for reviewing the manuscript during the busy schedule and providing valuable comments.

All the changes in the article are marked with bold and underline (E.g. changes).

Point 1: - line 13 - in the reviewer’s opinion instead off “recycled concrete block hollow” should be “recycled concrete hollow block”. This is stated in the article title and keywords, the abbreviation RCHB corresponds to that name, it is linguistically more correct.

Response 1: The "recycled concrete block hollow" had been changed to "recycled concrete hollow block" as recommended by the reviewer.

Point 2: - line 95 - in the reviewer’s opinion instead off “Experimental programs” will be better “Experimental program”.

Response 2: The "Experimental programs" had been changed to "Experimental program" as recommended by the reviewer.

Point 3: - line 113 - in the reviewer’s opinion instead off “The dimension of recycled concrete block” will be better “The dimension of recycled concrete hollow block”.

Response 3: The "The dimension of recycled concrete block" had been changed to "The dimension of recycled concrete hollow block" as recommended by the reviewer.

Point 4: - line 126 - in the reviewer’s opinion instead off “Process of masonry walls” will be better “Process of construction of masonry wall specimens”.

Response 4: The "Process of masonry walls" had been changed to "Process of construction of masonry wall specimens" as recommended by the reviewer.

Point 5: - line 152 - in the reviewer’s opinion instead off “Test program” will be better “Test procedure”.

Response 5: The "Test program" had been changed to "Test procedure" as recommended by the reviewer.

Point 6: - line 240 - in the reviewer’s opinion instead off “Experiment analysis” will be better “Test analysis”.

Response 6: The "Experiment analysis" had been changed to "Test analysis" as recommended by the reviewer.

Point 7: - line 323, fig. 10b - why the range for horizontal axis ends with 25? Data is only on a part of the chart, the scale of the axis should be adjust (-20,15).

Response 7: According to the reviewer's opinion, the scale of the axis had been adjusted (-20, 15). The unmodified version and modified version are shown below:

Unmodified version:(Image location)

Modified version:(Image location)

Point 8: - line 383, fig. 12a - why the range for horizontal axis ends with 25? Data is only on a part of the chart, the scale of the axis should be adjust (-0,20).

Response 8: According to the reviewer's opinion, the scale of the axis had been adjusted (0, 20). The unmodified version and modified version are shown below:

Unmodified version:(Image location)

Modified version:(Image location)

Point 9: - line 383, fig. 12b - why the range for horizontal axis ends with 25? Data is only on a part of the chart, the scale of the axis should be adjust (-0,15).

Response 9: According to the reviewer's opinion, the scale of the axis had been adjusted (0, 15). The unmodified version and modified version are shown below:

Unmodified version:(Image location)

Modified version:(Image location)
